# *Armeria maritima* (Mill.) Willd. Flower Hydromethanolic Extract for Cucurbitaceae Fungal Diseases Control

**DOI:** 10.3390/molecules28093730

**Published:** 2023-04-26

**Authors:** Eva Sánchez-Hernández, Pablo Martín-Ramos, Luis Manuel Navas-Gracia, Jesús Martín-Gil, Ana Garcés-Claver, Alejandro Flores-León, Vicente González-García

**Affiliations:** 1Department of Agricultural and Forestry Engineering, ETSIIAA, Universidad de Valladolid, Avda. Madrid 44, 34004 Palencia, Spain; pmr@uva.es (P.M.-R.); luismanuel.navas@uva.es (L.M.N.-G.); mgil@iaf.uva.es (J.M.-G.); 2Department of Plant Science, Agrifood Research and Technology Centre of Aragón, Instituto Agroalimentario de Aragón—IA2 (Universidad de Zaragoza-CITA), Avda. Montañana 930, 50059 Zaragoza, Spain; agarces@cita-aragon.es; 3Institute for the Preservation and Improvement of Valencian Agrodiversity, Universitat Politècnica de València, Camí de Vera, s/n, 46022 Valencia, Spain; alfloleo@doctor.upv.es; 4Department of Agricultural, Forestry, and Environmental Systems, Agrifood Research and Technology Centre of Aragón, Instituto Agroalimentario de Aragón—IA2 (Universidad de Zaragoza-CITA), Avda. Montañana 930, 50059 Zaragoza, Spain; vgonzalezg@aragon.es

**Keywords:** antifungal, Cucurbitaceae, *Fusarium*, GC–MS, halophyte, *Macrophomina*, *Neocosmospora*, phytocompounds, *Sclerotinia*

## Abstract

The cliff rose (*Armeria maritima*), like other halophytes, has a phenolics-based antioxidant system that allows it to grow in saline habitats. Provided that antioxidant properties are usually accompanied by antimicrobial activity, in this study we investigated the phytochemicals present in a hydromethanolic extract of *A. maritima* flowers and explored its antifungal potential. The main phytocompounds, identified by gas chromatography–mass spectrometry, were: hexadecanoic acid, octadecanoic acid, 9-octadecenoic acid, 3-(3,4-dihydroxy-phenyl)-acrylic acid ethyl ester, and benzeneacetaldehyde. The antifungal activity of the extract and its main constituents—alone and in combination with chitosan oligomers—was tested against six pathogenic taxa associated with soil-borne diseases of plant hosts in the family Cucurbitaceae: *Fusarium equiseti*, *F. oxysporum* f. sp. *niveum*, *Macrophomina phaseolina*, *Neocosmospora falciformis*, *N. keratoplastica*, and *Sclerotinia sclerotiorum*. In in vitro tests, EC_90_ effective concentrations in the 166−865 μg·mL^−1^ range were obtained for the chitosan oligomers–*A. maritima* extract conjugate complexes, lower than those obtained for fosetyl-Al and azoxystrobin synthetic fungicides tested for comparison purposes, and even outperforming mancozeb against *F. equiseti*. In ex situ tests against *S. sclerotiorum* conducted on artificially inoculated cucumber slices, full protection was achieved at a dose of 250 μg·mL^−1^. Thus, the reported results support the valorization of *A. maritima* as a source of biorationals for Cucurbitaceae pathogens protection, suitable for both organic and conventional agriculture.

## 1. Introduction

*Armeria maritima* (Mill.) Willd. (Plumbaginaceae), commonly known as sea thrift, sea rose, or cliff rose, is a compact, evergreen perennial plant that grows on cliffs and seashores in Iceland, the Atlantic coast of Europe, and the western region of the Baltic Sea [1].

*Armeria maritima* has been studied due to its potential for bioremediation, given its high tolerance to heavy metals [2]. Being a halophyte, *A. maritima* has a powerful antioxidant system based on phenolic acids and flavonoids [3,4]. Due to salinity, proline is the main amino acid [5]. Other bioactive compounds include *β*-alaninebetaine, glycinebetaine, and choline-O-sulphate [6]; gallic, caffeic, p-hydroxybenzoic as phenolic acids; and myricitrin, quercetin, and kaempferol glycoside flavonoids [3,4,7].

The dried flowering plant has antibiotic activity and has been used in traditional medicine to treat urinary infections, though it has been found to cause local irritation and dermatitis when used as a poultice. An *A. maritima* seed methanol extract has shown antibacterial activity against *Staphylococcus epidermidis* (Winslow and Winslow) [8]. However, there is a lack of information regarding the antimicrobial activities of extracts from other plant organs, which indicates a research gap.

Given that at present *A. maritima* is only valuable for horticultural and gardening purposes [9], its antimicrobial activity may offer an opportunity to valorize its extracts as biorationals for crop protection. The study presented herein has explored for the first time its potential to protect members of the family Cucurbitaceae, including watermelon (*Citrullus lanatus* (Thunb.) Matsum. & Nakai), melon (*Cucumis melo* L.), cucumber (*Cucumis sativus* L.), zucchini (*Cucurbita pepo* L.), and silver-seed gourd (*Cucurbita argyrosperma* Huber) [10], which rank among the top-ten economically important vegetable crops worldwide [11].

Cucurbits are prone to fungal diseases, which severely reduce crop production. *Fusarium* species, which cause wilt and root rot, are among the most destructive pathogens that affect these crops [12]. *Fusarium oxysporum* f. sp. *niveum* (E.F. Sm.) Snyder & H.N. Hansen causes wilt of watermelon [13], with race two being the most prevalent and damaging [14]. *Fusarium equiseti* (Corda) Sacc. affects cereal crop yield quality and quantity [15] and causes crown and root rot in cucurbits [16].

*Macrophomina phaseolina* (Tassi) Goid is a soil-borne pathogen that affects more than 500 plant species including melon, in which it is responsible for charcoal rot disease [17]. Symptoms include sunken and dark lesions at the base of the stem, as well as leaf and stem chlorosis, vine wilt, and stem and root rot [18]. Severely infected plants suffer from xylem flow disruption and fungal toxin damage, resulting in early death [19]. Controlling this disease is challenging: in melons, both fumigation and soil solarization have failed to eradicate it; and grafting melon scions onto *Cucurbita* spp. rootstocks—very effective in preventing plant collapse—involves additional costs, making it not profitable for growers [18].

The genus *Neocosmospora* (a taxon that belongs to the so-called *Fusarium solani* species complex, FSSC) contains saprobes, plant endophytes, and economically significant pathogens [20]. *Neocosmospora falciformis* (Carrión) Summerb. & Schroers has been linked to decay in several plant species [21,22,23,24,25] and to wilt and root rot of muskmelon in Spain [26]. Another member of the complex is *Neocosmospora keratoplastica* Geiser, O’Donnell, Short & Zhang, which also affects cucurbits by causing rotting and root rot [27].

Another polyphagous pathogenic fungus, *Sclerotinia sclerotiorum* (Lib.) de Bary, causes stem rot or white mold in many commercially significant crops, including cucurbits [28], leading to significant economic losses. Furthermore, sclerotia production allows it to survive in infected tissues, crop leftovers, or soil for up to eight years [29].

While the most effective, environmentally friendly, and safe control method for the aforementioned diseases would be the use of resistant cultivars, there are currently no widely available resistant cultivars or germplasm resources against most of these fungi. This leads to the widespread use of chemical pesticides, which can result in the emergence of resistant strains and environmental pollution [30]. An alternative approach, aligned with Directive 2009/128/EC, involves the use of biologically derived substances or biorationals.

The aim of this work was two-fold: (i) to investigate the phytoconstituents of *A. maritima* flowers hydromethanolic extract; and (ii) to examine the antifungal activity of the extract and its main constituents, alone and combined with chitosan oligomers (COS), against the above-mentioned horticultural phytopathogens. To achieve this latter goal, in addition to in vitro mycelium growth inhibition assays, the protective action of the COS−*A. maritima* conjugate complex was explored for the sustainable postharvest control of white mold on cucumber artificially infected with *S. sclerotiorum*.

## 2. Results

### 2.1. Vibrational Spectroscopy Characterization

Table 1 provides a summary of the main absorption bands observed in the infrared spectra of flowers, stems, and roots of *A. maritima*. The identified functional groups are compatible with the presence of alkaloids, polyphenols, organic acid esters, and other phytoconstituents (elucidated by GC–MS).

### 2.2. GC–MS Characterization

Up to forty compounds were identified in the chromatogram of *A. maritima* flowers hydromethanolic extract (Figure 1, Table 2). The main eleven phytocompounds were hexadecanoic acid (18%), 9-octadecenoic acid (14%), octadecanoic acid (9%), 2,1,3-benzothiadiazole (8.5%), methyl *β*-D-glucopyranoside (5.8%), 3-(3,4-dihydroxy-phenyl)-acrylic acid ethyl ester (5.3%), benzeneacetaldehyde (4.5%), 3,3,6-trimethyl-1,5-heptadiene (4%), altrosan (2.8%), 2,3-dihydro-benzofuran (2.6%), and 2-methoxy-4-vinylphenol (2.4%). The chemical structures of the most abundant phytochemicals are shown in Figure 2.

### 2.3. Antifungal Activity of the Extract

#### 2.3.1. In Vitro Antifungal Activity

The results of the antifungal susceptibility tests are summarized in Figure 3. For all the products assayed, higher concentrations led to lower radial growth of the fungal mycelium, resulting in statistically significant differences. In all cases, COS inhibited mycelial growth at 1500 μg·mL^−1^; meanwhile, the hydromethanolic extract of flowers achieved full inhibition at concentrations ranging from 375 to 1500 μg·mL^−1^, depending on the fungal taxa tested. Comparatively, the main constituents of the extract, i.e., hexadecanoic acid, 9-octadecenoic acid, and octadecanoic acid, exhibited similar or better activity than the whole extract. The formation of conjugate complexes further enhanced antifungal activity; COS–*A. maritima* extract led to complete inhibition at concentrations in the 250–1000 μg·mL^−1^ range, whereas full inhibition occurred at concentrations in the 78.12–375, 78.12–250, and 70.31–375 μg·mL^−1^ range for COS–hexadecanoic acid, COS–9-octadecenoic acid, and COS–octadecanoic acid conjugate complexes, respectively. To quantify this improved activity, effective concentration values were first calculated (Table 3), followed by synergy factors (Table 4) determined using the Wadley method. As a result, synergism (i.e., SFs > 1) was detected in all cases.

The results of mycelial growth inhibition using three conventional synthetic fungicides chosen for comparison are presented in Table 5. The highest inhibition rates were observed for the dithiocarbamate fungicide (mancozeb), which fully inhibited the mycelial growth of all phytopathogens at one-tenth of the manufacturer’s recommended dose (that is, 150 μg·mL^−1^), apart from *F. equiseti*, which was not completely inhibited at 1500 μg·mL^−1^. The organophosphate fungicide (fosetyl-Al) led to full inhibition of all fungus taxa at the recommended dose (i.e., 2000 μg·mL^−1^), except for *F. equiseti* and *S. sclerotiorum*. The strobilurin fungicide (azoxystrobin) was the least effective, failing to fully inhibit the growth of all phytopathogens at the recommended dose (62,500 μg·mL^−1^), except for *N. keratoplastica*.

#### 2.3.2. Ex Situ Antifungal Activity

Given that the COS−*A. maritima* conjugate complex was the most active product according to the previous in vitro tests, it was further tested as a protective treatment against white mold on cucumber fruits cv. “Urano”. Three different concentrations, corresponding to the minimum inhibitory concentration (MIC), MIC×2, and MIC×4 (i.e., 250, 500, and 1000 µg·mL^−1^, respectively), were assayed. Results are shown in Figure 4. In the positive control (i.e., *S. sclerotiorum* artificially inoculated on cucumber slices treated only with bi-distilled water), slices were fully colonized by the mold on the fifth day after inoculation and sclerotia were produced on the seventh day. In contrast, full protection was observed for the treated slices even at the lowest concentration (250 µg·mL^−1^), with an inhibition rate of 100%. Upon comparison of the slices’ weight evolution (Table 6), significant differences (*p* < 0.0001) were detected for the between-subjects and within-subjects effects, i.e., both time and treatment had a significant impact on the slices’ weight. A much more marked weight decrease, as a result of tissue maceration, was observed for the positive control, with no statistically significant differences between the negative control and the treated samples.

## 3. Discussion

### 3.1. On the Phytochemical Profile Obtained by GC–MS

Considering that the chosen hydromethanolic extraction mixture also solubilizes polar compounds (non-volatile) that cannot be detected by GC–MS without derivatization of the extract, it is important to note that such prior derivatization was not done in the present work due to the associated drawbacks. These include making the procedural preparation steps longer and more expensive (which would decrease the economic viability of the crop protection treatments), increased complexity and length of the data acquisition process due to the potential for impurities and the uncertainty of conversion of compounds into derivatives, as well as the use of toxic reagents [31]. Additionally, the injection of non-volatile compounds may result in damage to the GC capillary column.

Hexadecanoic acid (palmitic acid) has been previously identified in plants such as *Equisetum arvense* L. [32], *Limonium binervosum* (G.E.Sm.) C.E. Salmon [33], *Hibiscus syriacus* L. [34], and *Kigelia africana* (Lam.) Benth. [35], as well as in algae such as *Turbinaria ornata* (Turner) Agardh [36] and *Amphiroa zonata* Yendo [37]. Despite the taxonomic relatedness of *A. maritima* with *L. binervosum* [33] (both belong to the Plumbaginaceae family), hexadecanoic acid was the sole shared phytochemical. Octadecanoic acid (stearic acid) has been identified, for instance, in *Moringa oleifera* Lam. seed oil [38]. The simultaneous presence of octadecanoic and hexadecanoic acid has been reported in *Justicia wynaadensis* Heyne [39], *Piper betle* L. [40], and *Rosa damascena* Mill. Both are saturated long-chain fatty acids with stronger antifungal activity than unsaturated fatty acids, making them particularly suitable for the control of phytopathogens such as *F. oxysporum* [41].

9-Octadecenoic acid (*trans*-oleic acid or elaidic acid) has been found in small amounts in pomegranates, peas, cabbage [42], *Foeniculum vulgare* Mill. [43], and *Landolphia owariensis* Beauv. [44]. In pot experiments conducted by Liu et al. [41], the mixture of palmitic and oleic acids was found to enhance the growth of tomato and cucumber seedlings.

3-(3,4-Dihydroxy-phenyl)-acrylic acid ethyl ester (or ethyl caffeate) has previously been isolated from *Elsholtzia densa* Benth., *Ilex latifolia* Thunb. ex A.Murray, and *Ipomoea batatas* (L.) Lam. in antioxidant activity-guided phytochemical studies [45,46].

Benzeneacetaldehyde, detected in the flowers of *Rhododendron* spp. [47] and in the leaves of *Cantium parviflorum* Lam. [48], has been reported to possess antioxidant [49] and antimicrobial [48] activities.

The presence in the GC–MS chromatogram (at Rt = 12.18 min) of 2,1,3-benzothiadiazole (BTD) in a significant percentage (8.49%) but with a low Qual (<55) is a striking finding. BTD is a synthetic product used as an agrochemical, whose presence should be tentatively attributed to contamination. Nevertheless, it has previously been identified in a higher percentage (12.26%) in the ethanolic extract of *Lawsonia inermis* L. [50], so a possible natural origin cannot be completely ruled out. BTD is a plant defense inducer that has been used for the protection of various agronomically important crops, such as rice, wheat, potato, and tomato [51].

Methyl *β*-D-glucopyranoside, also known as *β*-methyl-D-glucoside (MeG) or methyl hexopyranoside (5.8%, Qual 58), is an *O*-glycosyl compound that has been found as a major compound in the leaves of the alpine herb *Geum montanum* L. and other plants of the *Rosaceae* family [52], as well as in *Echinospartum horridum* (Vahl) Rothm. [53]. It has been suggested that, like other methylated molecules (i.e., methyl-inositols), it might be involved in tolerance to osmotic stress [52].

### 3.2. On the Antifungal Activity and Mode of Action

The antifungal activity of *A. maritima* extract should be mainly attributed to its major constituents, i.e., fatty acids [54], as corroborated by other studies on fatty acid-rich plant extracts. For instance, hexadecanoic, 9-octadecenoic, and octadecanoic acids were found to comprise 4.5, 17.6, and 4.1%, respectively, of *Rosa damascena* Mill. essential oil, which showed antimicrobial activity at concentrations below 1000 µg·mL^−1^ against numerous human fungal pathogens, including *Candida albicans* (C.P. Robin) Berkhout (MIC = 125 µg·mL^−1^), *Staphylococcus epidermidis* (Winslow and Winslow 1908) Evans 1916, *Streptococcus pyogenes* Rosenbach, *Shigella dysenteriae* (Shiga 1898) Castellani and Chalmers, *Pseudomonas aeruginosa* (Schroeter 1872) Migula, *Salmonella paratyphi*-A (Brion and Kaiser 1902) Castellani and Chalmers 1919, *Escherichia coli* (Migula) Castellani and Chalmers (MIC = 250 µg·mL^−1^), *Staphylococcus aureus* Rosenbach, *Bacillus subtilis* (Ehrenberg) Cohn, *Klebsiella pneumoniae* (Schroeter) Trevisan (MIC = 500 µg·mL^−1^), and *Aspergillus brasiliensis* Varga, Frisvad & Samson (MIC = 1000 µg·mL^−1^) [55].

Concerning antifungal activity against fungal pathogens, *Peganum harmala* L. seed oil, containing 23.1, 5.4 and 3.1% oleic, palmitic, and stearic acid, respectively, showed activity against *Fusarium oxysporum* f. sp. *melonis* Snyder & Hansen, *Fusarium oxysporum* f. sp. *niveum*, *Fusarium solani* f. sp. *cucurbitae* Snyder & Hansen, *Rhizoctonia solani* J.G. Kühn, *Macrophomina phaseolina*, *Pythium* sp., *Alternaria* sp., *Colletotrichum* sp., and *Monosporascus cannonballus* Pollack & Uecker [56]. Hexadecanoic acid, at a concentration of 3900 μmol/L (1000 μg·mL^−1^), was found to reduce the growth of *F. oxysporum* f. sp. *cucumerinum* J.H. Owen, 1956, and *F. oxysporum* f. sp. *lycopersici* (Sacc.) Snyder & Hansen by 40%, and 36%, respectively [41]. It also reduced the radial growth of *Fusarium oxysporum* Schltdl. and *Fusarium avenaceum* (Fr.) Sacc. at 40 µg·mL^−1^ [57], although its effect was reversible. Hexadecanoic acid obtained from *Annona muricata* L. leaves showed fungicidal activity against *Alternaria solani* Sorauer, 1896 (MIC = 10,000 µg·mL^−1^), *Aspergillus erythrocephalus* Berk. & M.A.Curtis (MIC = 10,000 µg·mL^−1^), and *Aspergillus fumigatus* Fresen. (MIC > 15,000 µg·mL^−1^), but not against *Penicillium chrysogenum* Thom [58].

The underlying mode of action of these fatty acids has been mainly studied in human pathogenic fungi, not specifically against phytopathogens [41]. Nevertheless, it has been suggested that it involves their insertion into fungal membrane lipid bilayers, compromising membrane integrity and leading to uncontrolled release of intracellular proteins and electrolytes, ultimately resulting in cytoplasmic disintegration of fungal cells [59]. Hydrostatic turgor pressure within the cell leading to disruption of the fungal membrane has also been suggested as a mechanism of fungicidal action [60]. Additionally, fatty acids have been found to inhibit topoisomerase I, an enzyme involved in DNA strand breakage and repair and topological changes necessary for cellular processes [61], as well as N-myristoyltransferase, resulting in inhibition of fungal growth [62].

Nonetheless, it is worth noting that other constituents not tested as individual compounds may also contribute to the antifungal activity (as discussed in Section 3.1, based on other studies reported in the literature) and that the presence of synergism between phytoconstituents cannot be discounted.

With regard to COS, its antifungal activity is well-established [63], and is thought to be due to its positive charge interacting with the negative charge of the fungal cell membrane. This interaction leads to increased cell permeability [64], resulting in a loss of intracellular components which disrupts the osmotic pressure and causes cell death [65]. COS can also alter chitin levels, leading to a weakened cell wall [66], and can generate ROS that damage biomolecules, triggering apoptosis and necrosis. Additionally, COS can interfere with DNA and RNA synthesis [67].

Concerning the enhanced activity upon the formation of conjugate complexes, without additional in-detail experiments on the mechanism of its action, only an educated guess can be made at this stage. The observed synergism may stem from an enhanced additive fungicidal activity per se or by simultaneous action on multiple fungal metabolic sites [68], but it may also be due to an increase in the solubility and bioavailability of the bioactive compounds present in the extract mediated by COS.

### 3.3. Efficacy Comparisons

#### 3.3.1. Comparison with Conventional Fungicides

Upon comparison with three conventional fungicides, the MIC values obtained for the COS-*A. maritima* conjugate complex against *F. oxysporum* f. sp. *niveum*, *M. phaseolina*, *N. falciformis*, *N. keratoplastica*, and *S. sclerotiorum* (500, 500, 1000, 750, and 250 µg·mL^−1^) were higher (i.e., it was less effective) than those obtained for mancozeb (150 µg·mL^−1^). Nonetheless, in the case of *F. equiseti*, the conjugate complex was more effective than the dithiocarbamate (500 vs. >1500 µg·mL^−1^, respectively). Concerning the organophosphate fungicide (fosetyl-Al), it showed lower activity than the conjugate complex, requiring concentrations of 2000 µg·mL^−1^ against *F. oxysporum* f. sp. *niveum*, *M. phaseolina*, *N. falciformis*, and *N. keratoplastica*, and even higher doses against *F. equiseti* and *S. sclerotiorum*. As for the strobilurin (azoxystrobin), its efficacy was much lower than that of COS-*A. maritima*, requiring concentrations of over 62,500 µg·mL^−1^.

In line with the rationale behind the use of synthetic fungicides in pairs (not only to help prevent resistance development but also to benefit from the enhanced efficacy resulting from different modes of action), the better performance of the natural product versus the conventional fungicides may be tentatively attributed to the complex mixture of compounds found in the plant extract, given that these compounds may act synergistically to produce a more potent antifungal effect than synthetic fungicides based on one molecule.

#### 3.3.2. Comparison with Other Extracts Tested In Vitro against the Phytopathogens under Study

Table 7 presents a comparison of the efficacies reported for plant extracts and essential oils against five of the six studied phytopathogens. However, it should be noted that there are no data available for *N. falciformis*. It is important to exercise caution when interpreting these results, as the sensitivity may vary depending on the isolate. For instance, values for *F. oxysporum* spp. are presented due to the absence of specific data for *F. oxysporum* f. sp. *niveum*. Additionally, the results may be expressed in different forms (MIC values, inhibition rates, inhibition zones, etc.).

The non-conjugated *A. maritima* extract exhibited MIC values (1000, 750, 750, 1000, and 375 µg·mL^−1^) that are among the lowest for extracts. However, it is worth noting that some essential oils showed better performance. For *F. equiseti*, only *Plumbago zeylanica* L. root and *Tamarix gallica* L. bark extracts were more effective. Against *F. oxysporum* spp. and *S. sclerotiorum*, only *Cestrum nocturnum* L. flower extracts demonstrated activity comparable to that of the extract of *A. maritima*. Against *M. phaseolina*, it was only outperformed by *Oxalis corniculata* L., *P. zeylanica*, and *Antigonon leptopus* Hook. & Arn. extracts. Against *N. keratoplastica*, the efficacy of *A. maritima* extract was comparable to that of essential oils.

#### 3.3.3. Comparison with Other Extracts Tested Ex Situ for Cucumber Protection

There is a limited amount of research that has investigated the use of natural extracts to inhibit white mold on cucumber ex situ. In particular, extracts of *Cornus mas* L. (fruits or leaves), *Morus alba* L. (immature fruits or leaves), and *Prunus laurocerasus* L. (leaves) at 1000 mg·mL^−1^ were shown to arrest the development of *S. sclerotiorum* on cucumber, with inhibition percentages in the 94 to 100% range [96]. Another study conducted by the same group [97] found that chitosan at 2000 μg·mL^−1^ was also effective in protecting cucumber fruits against *S. sclerotiorum* lesions. In comparison with the aforementioned treatments, the efficacy of the COS–*A. maritima* conjugate complex was notably higher.

### 3.4. Limitations of the Study and Further Research

While the preliminary in vitro and ex situ results suggest that the proposed COS–*A. maritima* conjugate complexes have potential as antifungal agents against Cucurbitaceae fungal pathogens, further research is needed to assess their practical applicability for crop protection. Tests with different fungal strains would be required to factor in differences in sensitivity, and field tests should be conducted on various Cucurbitaceae species. Furthermore, the impact of the treatment on other Cucurbitaceae bacterial and fungal pathogens not tested in this study should also be taken into consideration if traditional fungicides are to be replaced with this alternative based on natural products. Additionally, the timing of application, dosage, and other practical aspects such as cost, degradation tolerance, and efficacy of long-term protection should also be carefully evaluated in future studies.

## 4. Materials and Methods

### 4.1. Plant Material and Chemicals

Specimens of *A. maritima* were collected in May 2021 in Cabo Ortegal, Cariño (Galicia, Spain); coordinates 43°46′12.1″ N 7°52′09.2″ W. They were identified and authenticated by Prof. Dr. Baudilio Herrero Villacorta (Departamento de Ciencias Agroforestales, ETSIIAA, Universidad de Valladolid) and voucher specimens are available at the herbarium of the ETSIIAA (code 17052021). To obtain a representative composite sample, plant parts from different specimens (n = 25) were mixed. The plant samples were dried in the shade and pulverized in a mechanical grinder to obtain a fine powder.

Hexadecanoic acid (CAS No. 57-10-3), 9-octadecenoic acid (CAS No. 112-80-1), and octadecanoic acid (CAS No. 57-11-4) were supplied by Alfa-Aesar (Haverhill, MA, USA). Tween^®^ 20 (CAS No. 9005-64-5) was acquired from Sigma Aldrich Química S.A. (Madrid, Spain). High-molecular weight chitosan (CAS No. 9012-76-4; MW: 310–375 kDa) was obtained from Hangzhou Simit Chem. & Tech. Co. (Hangzhou, China). The Neutrase^TM^ 0.8 L enzyme was supplied by Novozymes A/S (Bagsværd, Denmark). Potato Dextrose Agar (PDA) was purchased from Becton, Dickinson, and Company (Franklin Lakes, NJ, USA).

For comparison purposes, three commercial fungicides were used: Ortiva^®^ (azoxystrobin 25%; Syngenta, Basel, Switzerland), Vondozeb^®^ (mancozeb 75%; UPL Iberia, Barcelona, Spain), and Fesil^®^ (fosetyl-Al 80; Bayer, Leverkusen, Germany). These fungicides were provided by the Plant Health and Certification Center (CSCV) of the Gobierno de Aragón.

### 4.2. Phytopathogen Isolates

*F. equiseti* (MYC-1403), *F. oxysporum* f. sp. *niveum* (MYC-219), *M. phaseolina* (MYC-1178), *N. falciformis* (MYC-1345), *N. keratoplastica* (MYC-1250), and *S. sclerotiorum* (MYC-799) were supplied by the Mycology Lab of the Center for Research and Agrifood Technology of Aragón (CITA, Zaragoza, Spain) as subcultures on PDA.

### 4.3. Preparation of Armeria Extract, Chitosan Oligomers, and Conjugate Complexes

The flower samples were mixed (1:20 *w*/*v*) with a methanol/water solution (1:1 *v*/*v*) and heated in a water bath at 50 °C for 30 min, followed by sonication for 5 min in pulse mode with a 1-min stop every 2.5 min, using a model UIP1000 hdT probe-type ultrasonicator from Hielscher Ultrasonics (Teltow, Germany). The solution was then centrifuged at 9000 rpm for 15 min and the supernatant was filtered through Whatman No. 1 paper. For subsequent GC–MS analysis, 25 mg of the obtained freeze-dried extracts were dissolved in 5 mL of HPLC-grade MeOH to obtain a 5 mg·mL^−1^ solution, which was further filtered.

Chitosan oligomers were prepared according to the procedure previously described by our group [98], yielding a solution with a pH ranging from 4 to 6, containing oligomers of molecular weight less than 2 kDa.

The COS–*A. maritima* extract and COS−main bioactive compounds conjugate complexes were obtained by mixing the respective solutions in a 1:1 (*v/v*) ratio, followed by sonication for 15 min in 5 3-min pulses (so that the temperature did not exceed 60 °C). Attenuated total reflectance-Fourier transform infrared (ATR-FTIR) spectroscopy of the freeze-dried products was used to confirm the formation of the conjugate complexes.

### 4.4. Physicochemical Characterization

A Nicolet iS50 Fourier-transform infrared spectrometer from Thermo Scientific (Waltham, MA, USA) with an in-built diamond attenuated total reflection (ATR) system was utilized to collect the infrared vibrational spectra of plant organs. The spectra were registered between 400 and 4000 cm^−1^, with a spectral resolution of 1 cm^−1^, co-adding 64 scans.

A gas chromatograph model 7890A coupled to a quadrupole mass spectrometer model 5975C (both from Agilent Technologies, Santa Clara, CA, USA) was used to elucidate the constituents of *A. maritima* flowers hydromethanolic extract by gas chromatography-mass spectrometry (GC–MS). This characterization was outsourced to the research support services (STI) of the Universidad de Alicante (Alicante, Spain). The chromatographic conditions were: injection volume = 1 µL; injector temperature = 280 °C, in splitless mode; initial oven temperature = 60 °C, held for 2 min, followed by a ramp of 10 °C·min^−1^ up to a final temperature of 300 °C, held for 15 min. An HP-5MS UI chromatographic column (30 m length, 0.250 mm diameter, 0.25 µm film), also from Agilent Technologies, was employed for the separation of the compounds. The mass spectrometer conditions were: temperature of the electron impact source of the mass spectrometer = 230 °C and the quadrupole = 150 °C; ionization energy = 70 eV. The identification of components was based on a comparison of their mass spectra and retention times with those of authentic compounds and by computer matching with the database of the National Institute of Standards and Technology (NIST11).

### 4.5. In Vitro Antifungal Activity Assessment

The antifungal activity of the various treatments (including COS, the *A. maritima* flower extract, its main constituents (hexadecanoic acid, 9-octadecenoic acid, and octadecanoic acid), the conjugate complexes of all of them with COS, and certain commercial synthetic fungicides) was determined using the agar dilution method as per the EUCAST antifungal susceptibility testing standard procedures [99]. Stock solution aliquots were incorporated into the pouring PDA medium to produce final concentrations ranging from 15.62 to 1500 μg·mL^−1^. Mycelial plugs (⌀ = 5 mm), from the margin of 1-week-old PDA cultures of *F. equiseti*, *F. oxysporum* f. sp. *niveum*, *M. phaseolina*, *N. falciformis*, *N. keratoplastica*, and *S. sclerotiorum* were transferred to the center of PDA plates prepared with the aforementioned concentrations (3 plates per treatment and concentration, with 2 duplicates). The plates were incubated at 25 °C in the dark for 1 week. The control consisted in replacing the extract with the solvent used for extraction (i.e., methanol:water 1:1 *v/v*) in the PDA medium. Inhibition of mycelial growth was estimated according to the formula ((*d_c_ − d_t_*)/*d_c_*) × 100, where *d_c_* and *d_t_* represent the mean diameters of the control and treated fungal colonies, respectively. Given that the homogeneity and homoscedasticity requirements were met (according to Shapiro–Wilk and Levene tests, respectively), the results of mycelial growth inhibition were statistically analyzed in IBM SPSS Statistics (IBM, New York, NY, USA) v.25 software using one-way analysis of variance (ANOVA), followed by post hoc comparison of means using Tukey’s test at *p* < 0.05.

Effective concentrations (EC_50_ and EC_90_) were determined via PROBIT analysis in IBM SPSS Statistics v.25. Interaction levels, i.e., synergy factors (SF), were estimated according to the Wadley method [100], which is based on the notion that one component of the mixture can substitute at a constant proportion for the other. Therefore, the anticipated efficacy of the mixture can be directly determined from the efficacy of the constituents when the relative proportions are known (as is the case here). SF = 1 indicates similar joint action (i.e., additivity), SF > 1 implies synergistic action, and SF < 1 implies antagonistic action between the two fungicide products.

### 4.6. Post-Harvest Protection Test in Cucumber

The cucumber fruits (*C. sativus* cv. “Urano”) used to ascertain the ex situ protective effect of COS−*A. maritima* conjugate complex against *S. sclerotiorum* were sourced from the ‘Huerta de Carabaña’ orchard (Carabaña, Madrid, Spain) and previously grown under organic farming standards, without the use of synthetic pesticides. To begin the experiments within 24 h of harvest, the fruits were picked and sent by refrigerated express courier service. During selection, the fruits were chosen for their firmness, consistent size, caliber, lack of physical damage, and absence of signs of bacterial or fungal infection.

In controlled laboratory conditions, the efficacy of the treatment was determined by artificial inoculation of cucumber slices. The procedure was slightly modified from that proposed by Onaran and Yanar [96] and described in Sánchez-Hernández et al. [101]. The cucumber fruits were initially disinfected with a 3% NaOCl solution for 2 min, washed 3 times with sterile distilled water and dried in a laminar-flow hood on sterile absorbent paper. Then, under sterile conditions, cucumber fruits were cut into 8 mm-thick slices with a sterile knife. In each Petri plate containing sterile filter paper, one cucumber slice was placed, and a superficial wound (ø = 3 mm) was created in the equatorial zone of each slice. In these wounds, 100 µL of the COS−*A. maritima* conjugate complex at three concentrations (at the MIC obtained in previous in vitro assays, at MIC×2, and at MIC×4, i.e., 250, 500, and 1000 μg·mL^−1^, respectively) were applied, followed by a two-hour waiting period for complete absorption. Then, a plug of *S. sclerotiorum* PDA culture was placed in each wound (with the mycelium facing the fruit wound). In the negative control, wounds were treated only with distilled water (without the pathogen), while positive controls were treated with distilled water and inoculated with the pathogen. All cucumber slices were incubated at 22 ± 2 °C and 75–90% RH for 7 days. Cucumber slices were weighed daily to study weight loss and disease progression, with the weight of each slice on day 0 being set as 100%. The experiment was conducted using three replicates, repeated three times. The results were statistically analyzed by repeated measures ANOVA with post hoc comparison of means by Tukey’s test.

## 5. Conclusions

The application of GC–MS to an hydromethanolic extract of *A. maritima* flowers identified hexadecanoic acid (18%), 9-octadecenoic acid (14%), and octadecanoic acid (9%) as its main phytoconstituents. Subsequent antifungal tests against *F. equiseti*, *F. oxysporum* f. sp. *niveum*, *N. falciformis*, *N. keratoplastica*, *M. phaseolina*, and *S. sclerotiorum* revealed that the extract had strong inhibitory effects, with MIC values ranging from 375 to 1500 μg·mL^−1^. This activity was even more prominent after conjugation with chitosan oligomers, resulting in MICs between 250 and 1000 μg·mL^−1^ depending on the fungal taxa. In comparison, these inhibitory effects were greater than those of conventional chemicals such as fosetyl-Al and azoxystrobin and, in the case of *F. equiseti*, exceeded those of mancozeb. The conjugate complex was also tested as a protective treatment in ex situ experiments on cucumber slices artificially inoculated with *S. sclerotiorum*, showing full inhibition at a concentration of 250 μg·mL^−1^. The results suggest that the extracts of this halophyte could be valorized as biorationals for the protection of cucurbits against certain soil-borne diseases. However, further studies are needed to assess the impact of the proposed treatment on other Cucurbitaceae pathogens and long-term protection. Additionally, practical aspects for its field application need to be optimized.

## Figures and Tables

**Figure 1 molecules-28-03730-f001:**
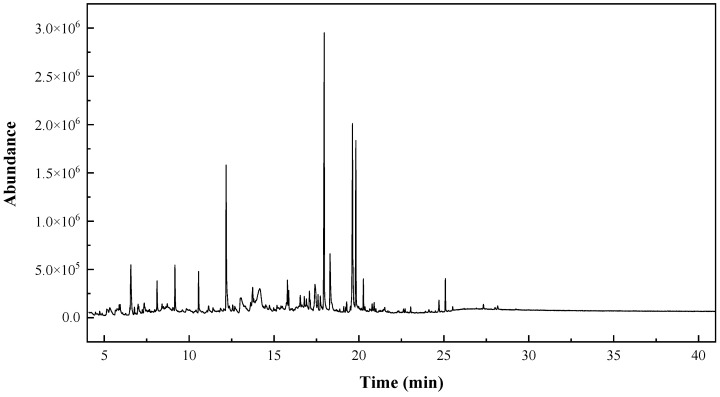
GC–MS chromatogram of *A. maritima* flower extract.

**Figure 2 molecules-28-03730-f002:**
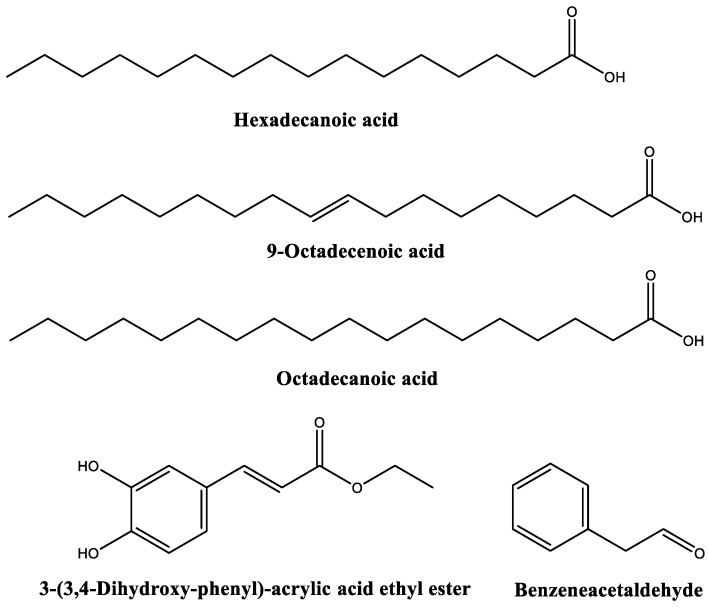
Main phytochemicals identified in *A. maritima* flower extract (with Qual > 90).

**Figure 3 molecules-28-03730-f003:**
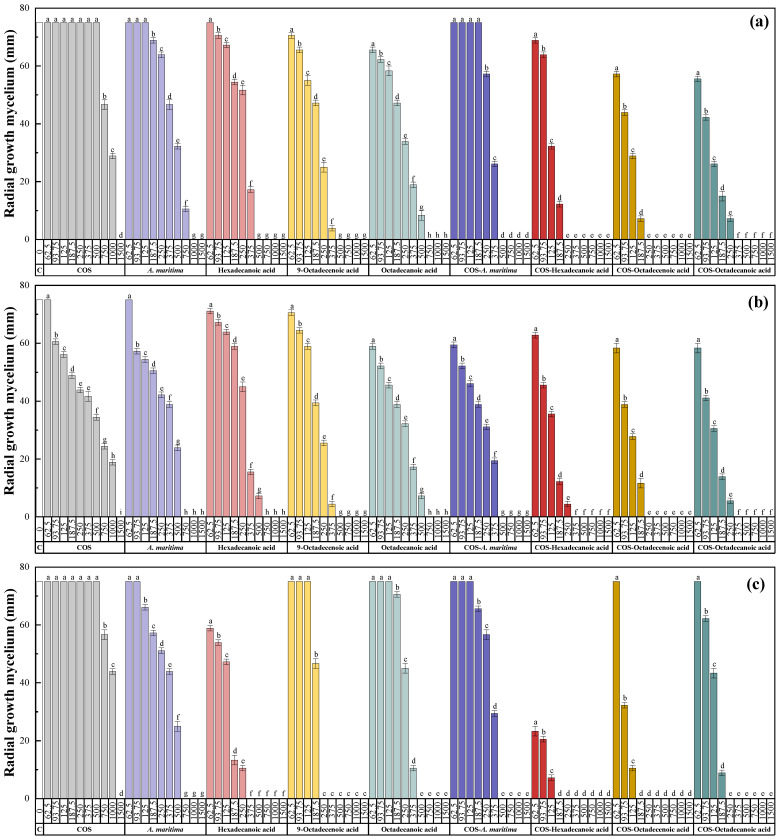
Inhibition of the radial growth of the mycelium of (**a**) *F. equiseti*, (**b**) *F. oxysporum* f. sp. *niveum*, (**c**) *M. phaseolina*, (**d**) *N. falciformis*, (**e**) *N. keratoplastica*, and (**f**) *S. sclerotiorum* in in vitro tests performed with PDA medium amended with different concentrations (in the 15.62–1500 µg·mL^−1^ range) of chitosan oligomers (COS), *A. maritima* flower extract, its main phytochemical constituents (viz., hexadecanoic acid, 9-octadecenoic acid, and octadecanoic acid), and their respective conjugated complexes. C (white bars) represents the controls. The efficacies of the concentrations labeled with the same letters are not statistically different at *p* < 0.05. Standard deviations are represented by error bars.

**Figure 4 molecules-28-03730-f004:**
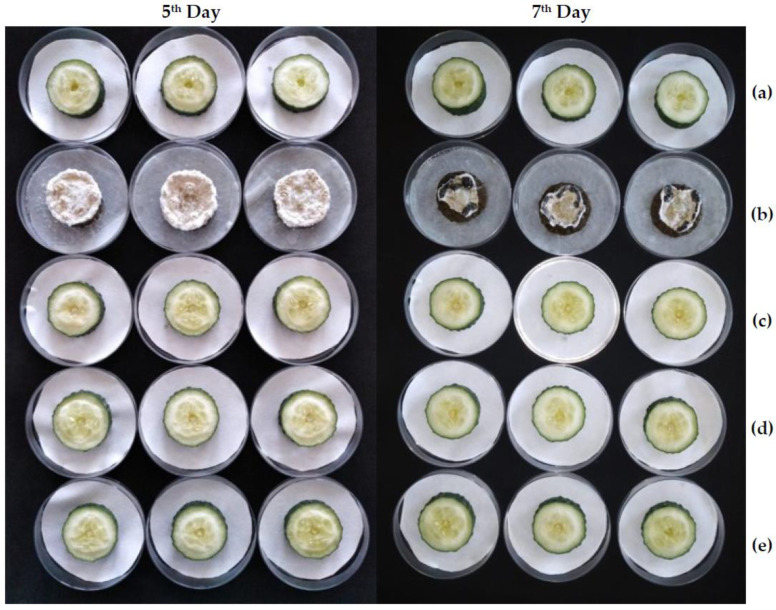
White mold decay symptoms on cucumber slices 5 and 7 days after artificial inoculation: (**a**) negative control; (**b**) slices artificially inoculated with *S. sclerotiorum* (positive control); slices treated with the COS−*A. maritima* conjugate complex at different concentrations, namely (**c**) 250 µg·mL^−1^, (**d**) 500 µg·mL^−1^, and (**e**) 1000 µg·mL^−1^, and subsequently inoculated with *S. sclerotiorum*.

**Table 1 molecules-28-03730-t001:** Main absorption bands in the infrared spectra of *Armeria maritima* plant organs. Wavenumbers are expressed in cm^−1^.

Flowers	Root	Stem	Assignment
3290	3282	3355	OH group in phenolic compounds
2918	2921	2919	O−H stretching
2850	2851	2850	–CH_2_ symmetric stretching (cutine and wax); CH_2_–(C6)– bending (cellulose)
1732		1726	C=O stretching of alkyl ester
1651			C=O (amide I)
1633	1620	1639	skeletal vibration due to aromatic C=C ring stretching and C=O stretching
1605			C=C stretching
	1545	1546	aromatic C=C stretching
1515		1517	aromatic skeletal
14,351,416	1445	14,431,414	symmetric aromatic ring stretching vibration (C=C ring);aromatic skeletal combined with C−H in-plane deformation and stretching
1367	1344	1371	aliphatic C−H stretching in methyl and phenol OH
1308		1321	C−H vibration of the methyl group
1240	1238	1236	aromatic ring−O−aromatic ring stretching
1201			present in hemicelluloses
1162	1145	1152	C-O-C asymmetric stretching in cellulose I and cellulose II
		1103	in-plane =C−H bending/C=C stretching
1030	1034	1033	C–O stretching/O−H out plane bending
896	919		*β*-glycosidic linkages (glucose units of cellulose chains)

**Table 2 molecules-28-03730-t002:** Main phytoconstituents identified in *A. maritima* flower extract.

RT (min)	Peak Area (%)	Assignment	Qual
5.3273	1.6888	2-Furancarboxaldehyde, 5-methyl-	93
6.5084	0.3951	Piperazine, 1,4-dimethyl-	52
6.5618	4.4740	Benzeneacetaldehyde	93
6.7815	0.3599	2,5-Dimethyl-4-hydroxy-3(2H)-furanone	62
6.9951	0.3635	Thiazole	43
7.3572	0.4867	Cyclopropanecarboxylic acid, 1-amino-	59
8.1110	1.7736	4H-Pyran-4-one, 2,3-dihydro-3,5-dihydroxy-6-methyl-	91
8.4077	0.2368	Benzoic acid	55
9.1615	2.5799	Benzofuran, 2,3-dihydro-	83
10.5563	2.3649	2-Methoxy-4-vinylphenol	95
11.1439	0.6006	Methyl 3-methoxyamino-propanoate	38
12.1826	8.4919	2,1,3-Benzothiadiazole/2-trifluoromethyl imidazole	53
12.2538	0.7550	2,2′-Bipyridine	92
13.0669	2.7909	3,4-Altrosan	49
13.6189	0.1442	1-Pyrrolidinyloxy, 3-amino-2,2,5,5-tetramethyl-	53
13.7376	1.3464	3-Hydroxy-4-methoxybenzoic acid	95
13.8029	0.2216	3-Piperidinone, 1,6-dimethyl-	64
14.1590	5.8153	*β*-D-Glucopyranoside, methyl	58
15.7378	0.5604	4-((1E)-3-Hydroxy-1-propenyl)-2-methoxyphenol	46
15.7912	1.5928	2-Propenoic acid, 3-(4-hydroxyphenyl)-, methyl ester	98
15.8625	1.0835	Tetradecanoic acid	98
16.5450	0.8155	Benzoic acid, 4-hydroxy-3,5-dimethoxy-	98
16.7884	0.5167	2-Propenoic, 3-(4-hydroxy-3-methoxyphenyl)-, methyl ester	99
16.9071	0.5716	Pentadecanoic acid	96
17.0851	1.0764	2-Propenoic acid, 3-(4-hydroxy-3-methoxyphenyl)-	94
17.1326	0.4642	2-Tetradecene, (E)-	90
17.4175	4.0260	1,5-Heptadiene, 3,3,6-trimethyl-	38
17.5896	0.9559	Pentadecanoic acid, 14-methyl-, methyl ester	97
17.7262	1.1349	5-Undecene	46
17.9576	18.0487	n-Hexadecanoic acid (or palmitic acid)	99
18.3019	5.3442	3-(3,4-Dihydroxy-phenyl)-acrylic acid ethyl ester	91
19.2753	0.5834	11-Octadecenoic acid, methyl ester	99
19.6195	14.4270	9-Octadecenoic acid, (E)-//Oleic acid	99
19.8154	9.0166	Octadecanoic acid (or stearic acid)	99
20.2605	1.6119	4-Methoxybenzoic acid, 2,4,5-trichlorophenyl ester	43
20.7769	0.4864	7-Butyl-3,4,5,6(2H)-tetrahydroazepine	49
20.8956	0.5923	Isophthalic acid, di(but-3-yn-2-yl) ester	35
24.7179	0.6977	Octabenzone	98
25.0919	1.5047	Supraene	98

**Table 3 molecules-28-03730-t003:** Effective concentrations (expressed in µg·mL^−1^) against *F. equiseti*, *F. oxysporum* f. sp. *niveum*, *M. phaseolina*, *N. falciformis*, *N. keratoplastica*, and *S. sclerotiorum* of chitosan oligomers (COS), *A. maritima* flower extract, its main phytochemical constituents, and their respective conjugate complexes.

Treatment	EC	*F. equiseti*	*F. oxysporum* f. sp. *niveum*	*M. phaseolina*	*N. falciformis*	*N. keratoplastica*	*S. sclerotiorum*
COS	EC_50_	867.8	455.9	1151.7	721.8	677.5	864.3
	EC_90_	1350.4	1296.4	1420.5	1130.2	1295.4	1344.8
*A. maritima* flower extract	EC_50_	448.0	387.4	413.2	463.4	482.2	13.5
	EC_90_	832.4	660.1	664.2	1053.1	845.1	235.6
Hexadecanoic acid	EC_50_	297.1	275.9	156.0	268.3	230.0	120.3
	EC_90_	422.8	472.8	278.5	501.8	346.5	164.0
9-octadecenoic acid	EC_50_	213.7	195.8	213.8	111.7	46.8	62.8
	EC_90_	347.2	354.3	238.7	242.0	163.2	110.0
Octadecanoic acid	EC_50_	231.3	202.6	269.7	126.4	35.5	27.7
	EC_90_	552.6	503.3	385.7	462.6	214.5	137.2
COS–*A. maritima*	EC_50_	320.3	205.7	308.1	444.1	442.7	129.2
	EC_90_	461.5	452.4	482.5	865.2	683.4	165.9
COS– hexadecanoic acid	EC_50_	110.9	114.0	36.7	113.9	103.6	29.3
	EC_90_	210.8	224.6	136.4	245.8	168.7	61.5
COS– 9-octadecenoic acid	EC_50_	121.5	107.8	83.2	79.5	29.7	21.1
	EC_90_	199.5	218.9	127.8	91.0	74.8	62.4
COS– octadecanoic acid	EC_50_	109.3	102.8	131.1	86.8	9.3	25.4
	EC_90_	256.3	231.4	193.1	101.9	48.9	61.2

**Table 4 molecules-28-03730-t004:** Synergy factors for conjugate complexes estimated according to the Wadley method.

Treatment	EC	*F. equiseti*	*F. oxysporum* f. sp. *niveum*	*M. phaseolina*	*N. falciformis*	*N. keratoplastica*	*S. sclerotiorum*
COS– *A. maritima*	EC_50_	1.84	2.04	1.97	1.27	1.27	2.34
	EC_90_	2.34	1.93	1.88	1.36	1.50	2.42
COS– hexadecanoic acid	EC_50_	3.99	3.02	7.49	3.43	3.31	7.21
	EC_90_	3.26	3.09	3.41	2.96	3.24	4.75
COS– 9-octadecenoic acid	EC_50_	2.82	2.54	4.33	2.43	2.95	5.55
	EC_90_	2.84	2.54	3.20	4.50	3.88	3.26
COS– octadecanoic acid	EC_50_	3.34	2.73	3.33	2.43	7.25	2.11
	EC_90_	3.17	3.13	3.14	4.50	7.53	4.07

**Table 5 molecules-28-03730-t005:** Radial growth of the mycelium of *F. equiseti*, *F. oxysporum* f. sp. *niveum*, *M. phaseolina*, *N. falciformis*, *N. keratoplastica*, and *S. sclerotiorum* in in vitro assays performed on a PDA medium with two concentrations (the manufacturer’s recommended dose and a tenth of the same) of three commercial synthetic fungicides.

Commercial Fungicide	Pathogen	Radial Growth of Mycelium (mm)			Inhibition (%)	
		Control (PDA)	Rd/10	Rd *	Rd/10	Rd *
Azoxystrobin	*F. equiseti*	75.0	50.0	46.7	33.3	37.8
	*F. oxysporum* f. sp. *niveum*	75.0	45.0	40.0	40.0	46.7
	*M. phaseolina*	75.0	38.3	16.7	48.9	77.8
	*N. falciformis*	75.0	43.3	28.3	42.2	62.2
	*N. keratoplastica*	75.0	10.0	0.0	86.7	100.0
	*S. sclerotiorum*	75.0	14.0	9.0	81.3	88.0
Mancozeb	*F. equiseti*	75.0	70.0	25.0	6.7	66.7
	*F. oxysporum* f. sp. *niveum*	75.0	0.0	0.0	100.0	100.0
	*M. phaseolina*	75.0	0.0	0.0	100.0	100.0
	*N. falciformis*	75.0	0.0	0.0	100.0	100.0
	*N. keratoplastica*	75.0	0.0	0.0	100.0	100.0
	*S. sclerotiorum*	75.0	0.0	0.0	100.0	100.0
Fosetyl-Al	*F. equiseti*	75.0	75.0	30.0	0.0	20.0
	*F. oxysporum* f. sp. *niveum*	75.0	66.7	0.0	11.1	100.0
	*M. phaseolina*	75.0	75.0	0.0	0.0	100.0
	*N. falciformis*	75.0	61.7	0.0	17.8	100.0
	*N. keratoplastica*	75.0	66.7	0.0	11.1	100.0
	*S. sclerotiorum*	75.0	75.0	13.3	0.0	82.2

* Rd stands for the recommended dose, i.e., 62.5 mg·mL^−1^ of azoxystrobin (250 g·L^−1^ for Ortiva^®^, azoxystrobin 25%), 1.5 mg·mL^−1^ of mancozeb (2 g·L^−1^ for Vondozeb^®^, mancozeb 75%), and 2 mg·mL^−1^ of fosetyl-Al (2.5 g·L^−1^ for Fosbel^®^, fosetyl-Al 80%). The radial growth of the mycelium for the control (PDA) was 75 mm. All mycelial growth values (in mm) are average values (*n* = 3).

**Table 6 molecules-28-03730-t006:** Evolution of the weights of cucumber slices for each treatment (normalized to the weight of the slices at the beginning of the experiment).

Treatment	Day 1	Day 2	Day 3	Day 4	Day 5	Day 6	Day 7
C−	1.01 ± 0.00 ^a^	1.00 ± 0.00 ^a^	0.99 ± 0.00 ^ab^	0.96 ± 0.00 ^ab^	0.91 ± 0.00 ^ab^	0.89 ± 0.01 ^a^	0.85 ± 0.01 ^a^
C+	1.01 ± 0.01 ^a^	1.00 ± 0.01 ^a^	0.93 ± 0.01 ^b^	0.91 ± 0.01 ^b^	0.87 ± 0.02 ^b^	0.59 ± 0.04 ^b^	0.43 ± 0.05 ^b^
MIC	1.01 ± 0.00 ^a^	1.00 ± 0.00 ^a^	0.99 ± 0.00 ^a^	0.96 ± 0.00 ^ab^	0.91 ± 0.01 ^ab^	0.90 ± 0.01 ^a^	0.84 ± 0.01 ^a^
MIC×2	1.02 ± 0.03 ^a^	1.01 ± 0.03 ^a^	1.00 ± 0.03 ^a^	0.97 ± 0.03 ^a^	0.92 ± 0.03 ^ab^	0.92 ± 0.03 ^a^	0.82 ± 0.10 ^a^
MIC×4	1.03 ± 0.04 ^a^	1.03 ± 0.04 ^a^	1.02 ± 0.04 ^a^	0.98 ± 0.04 ^a^	0.94 ± 0.04 ^a^	0.93 ± 0.04 ^a^	0.93 ± 0.08 ^a^

C− and C+ represent negative and positive controls, respectively. Means (*n* = 9) followed by a common letter are not significantly different by Tukey’s test at the 5% level of significance.

**Table 7 molecules-28-03730-t007:** Efficacy of plant extracts and essential oils reported in the literature against the phytopathogens under study.

Pathogen	Source/Extraction Medium	Plant	Efficacy	Ref.
*F. equiseti*	Aqueous ammonia	*Tamarix gallica* bark	MIC = 750 µg·mL^−1^	[69]
	Commercial essential oil	*Zataria multiflora*	MIC = 99–145 µg·mL^−1^	[70]
		*Heracleum persicum*	MIC = 795–1180 µg·mL^−1^	
		*Pinaceae*	MIC = 163–176 µg·mL^−1^	
		*Cuminum cyminum*	MIC = 75–99 µg·mL^−1^	
		*Foeniculum vulgare*	MIC = 63–69 µg·mL^−1^	
	Oil cake extracts at 1–3%	*Brassica napus*	IR = 43.6–59.1%	[71]
		*Cocos nucifera*	IR = 7.6–22.4%	
		*Sesame indicum*	IR = 49.4–56.1%	
		*Glycine max*	IR = 0.4–5.9%	
	Essential oil	*Piper auritum* aerial parts	MIC_50_ = 9000 µg·mL^−1^	[72]
	Ethanol extract	*Emblica officinalis* fruits	IZ = 9.5 mm	[73]
	Acetone extract		IZ = 10 mm	
	Ethanol extract	*Plumbago zeylanica* roots	MIC = 250 µg·mL^−1^	[74]
	Aqueous extract at 25%	*Acacia nilotica* leaves	IR = 67%	[75]
		*Achras zapota* leaves	IR = 44.8%	
		*Datura stramonium* leaves	IR = 87.3%	
		*E. officinalis* leaves	IR = 75.8%	
		*Eucalyptus globulus* leaves	IR = 62.0%	
		*Lawsonia inermis* leaves	IR = 78.3%	
		*Mimusops elengi* leaves	IR = 85.8%	
		*Peltophorum pterocarpum* leaves	IR = 74.3%	
		*Polyalthia longifolia* leaves	IR = 40.5%	
		*Prosopis juliflora* leaves	IR = 76.8%	
		*Punica granatum* leaves	IR = 77.5%	
		*Syzygium cumini* leaves	IR = 68.8%	
	Aqueous extract	*Filipendula* spp. flowers	IR = 100%	[76]
		*Allium sativum*	IR = 92.2%	
*F. oxysporum* spp.	Aqueous extract at 5, 10, and 20%	*Azadirachta indica* leaves	n.a.	[77]
		*Parthenium hysterophorus* leaves + flowers	IR = 2.6–15.9%	
		*Momordica charantia* leaves	IR = 14.4–24.4%	
		*A. sativum* cloves	IR = 52.6–63.3%	
		*Eucalyptus globules* leaves	IR = 34.3–61.8%	
		*Calotropis procera* leaves	n.a.	
		*Aloe vera* leaves	IR = 16.6%	
		*Beta vulgaris* root	IR = 6.3–10.3%	
		*D. stramonium* leaves	IR = 61.1%	
	Aqueous extract at 1%	*P. granatum* fruits	IR = 78%	[78]
	Propanol extract at 1%		IR = 62%	
	Hexane extract	*Cestrum nocturnum* flowers	MIC = 1000 µg·mL^−1^	[79]
	Chloroform extract		MIC = 1000 µg·mL^−1^	
	Ethyl acetate extract		MIC = 500 µg·mL^−1^	
	Methanol extract		MIC = 500 µg·mL^−1^	
	Crude extract at 5, 10, and 20%	*A. indica* leaves	IR = 24.1–62.0%	[80]
		*Ocimum sanctum* leaves	IR = 7.0–17.0%	
		*Datura metel* leaves	IR = 10.1–34.2%	
		*Cassia alata* leaves	IR = 46.8–74.7%	
		*Asparagus racemosus* roots	IR = 44.3–57.0%	
		*A. sativum* bulbs	IR = 17.6–34.2%	
		*Zingiber officinale* tubers	IR = 23.7–39.5%	
	Ethanol extract	*Flourensia microphylla* leaves	MIC = 1500 µL·L^−1^	[81]
		*F. cernua* leaves	MIC = 1500 µL·L^−1^	
		*F. retinophylla* leaves	MIC = 1500 µL·L^−1^	
	Aqueous extract at 5–50%	*Moringa oleifera* leaves	IR = 43.4–100%	[82]
		*M. oleifera* roots	IR = 48.8–100%	
		*M. oleifera* pud coats	IR = 36–100%	
	Commercial essential oil	*Z. multiflora*	MIC = 77–183 µg·mL^−1^	[70]
		*H. persicum*	MIC = 753–2250 µg·mL^−1^	
		*Pinaceae*	MIC = 113–147 µg·mL^−1^	
		*C. cyminum*	MIC = 70–145 µg·mL^−1^	
		*F. vulgare*	MIC = 77–94 µg·mL^−1^	
	Essential oil	*P. auritum* aerial parts	MIC_50_ = 6000–9000 µg·mL^−1^	[72]
	Aqueous extract at 25%	*A. nilotica* leaves	IR = 82%	[75]
		*A. zapota* leaves	IR = 34.8%	
		*D. stramonium* leaves	IR = 67.5%	
		*E. officinalis* leaves	IR = 79.5%	
		*E. globulus* leaves	IR = 59.3%	
		*L. inermis* leaves	IR = 82.0%	
		*M. elengi* leaves	IR = 86.0%	
		*P. pterocarpum* leaves	IR = 53.3%	
		*P. longifolia* leaves	IR = 36.3%	
		*P. juliflora* leaves	IR = 80.3%	
		*P. granatum* leaves	IR = 73.8%	
		*S. cumini* leaves	IR = 69.5%	
	Aqueous extract	*Filipendula* spp. flowers	IR = 95.9%	[76]
		*A. sativum*	IR = 81.4%	
	Ethanolic extract	*Mentha spicata*	MIC = 5%	[83]
	Aqueous extract	*A. sativum* leaves	MIC = 7000 µg·mL^−1^	[84]
*M. phaseolina*	Aqueous extract at 5, 10, and 20%	*A. indica* leaves	n.a.	[77]
		*P. hysterophorus* leaves + flowers	n.a.	
		*M. charantia* leaves	n.a.	
		*A. sativum* cloves	IR = 100%	
		*E. globules* leaves	n.a.	
		*C. procera* leaves	n.a.	
		*A. vera* leaves	n.a.	
		*B. vulgaris* root	n.a.	
		*D. stramonium* leaves	IR = n.a –57.7%	
	Aqueous extract at 5–50%	*M. oleifera* leaves	IR = 17.8–82.2%	[82]
		*M. oleifera* roots	IR = 20–87.4%	
		*M. olifera* pud coats	IR = 13.8–82.2%	
	Chloroform extract	*Ageratum conyzoides* leaves	n.a.	[85]
		*Antigonon leptopus* leaves		
		*Chromolaena odorata* leaves		
		*Oxalis corniculata* leaves		
		*Passiflora foetida* leaves		
	Methanol extract	*A. conyzoides* leaves	MIC = 1250 µg·mL^−1^	
		*A. leptopus* leaves	MIC = 625 µg·mL^−1^	
		*C. odorata* leaves	MIC = 2500 µg·mL^−1^	
		*O. corniculata* leaves	MIC = 78 µg·mL^−1^	
		*P. foetida* leaves	MIC = 1250 µg·mL^−1^	
	Aqueous extract at 5–20%	*Citrus aurantifolia* leaves	IR = 75.6–96.7%	[86]
	Ethanol extract	*E. officinalis* fruits	n.a.	[73]
	Acetone extract			
	Ethanol extract	*P. zeylanica* roots	MIC = 500 µg·mL^−1^	[74]
*N. keratoplastica*	Essential oil	*Trachyspermum ammi* seeds	n.a.	[87]
	Essential oil	*Kaempferia parviflora* rhizome	IZ = 17–18 mm	[88]
	Essential oil	*Pogostemon cablin* flowers + leaves	n.a. at 500 µg·mL^−1^	[89]
	Essential oil	*Origanum vulgare* subsp. *hirtum*	MIC = 800 µg·mL^−1^	[90]
*S. sclerotiorum*	Hexane extract	*C. nocturnum* flowers	MIC = 1000 µg·mL^−1^	[79]
	Chloroform extract		MIC = 500 µg·mL^−1^	
	Ethyl acetate extract		MIC = 250 µg·mL^−1^	
	Methanol extract		MIC = 500 µg·mL^−1^	
	Essential oils at 1, 2.5, and 5%	*Thymus vulgaris*	n.a.	[91]
		*Nigella sativa*	n.a.	
		*Origanum majorana*	MIC = 2.5%	
		*Syzygium aromaticum*	MIC = 2.5%	
		*Salvia rosmarinus*	n.a.	
	Essential oils at 20%	*Ocimum basilicum*	IR = 4.1%	[92]
		*A. sativum*	IR = 28.2%	
		*Cymbopogon citratus*	IR = 9.1%	
		*Nerium oleander*	IR = 14.1%	
		*A. indica*	IR = 35.5%	
		*Allium cepa*	IR = 16.9%	
	Essential oil	*Z. officinale*	MIC = 1000 µg·mL^−1^	[93]
	Aqueous extracts	*Trachystemon orientalis* leaves	MIC = 7%	[94]
		*T. orientalis* flowers	MIC = 1%	
	Crude extracts	*Rosmarinus officinalis* leaves	MIC = 10%	[95]
		*Salvia fructicosa* leaves	MIC = 20%	
	Ethanol extract	*M. spicata*	MIC = 5%	[83]
	Aqueous extract	*A. sativum* leaves	MIC = 5000 µg·mL^−1^	[84]

IR: inhibition rate; IZ: inhibition zone; MIC: minimum inhibitory concentration; MIC_50_: minimum inhibitory concentration that inhibited 50% of the radial growth; n.a.: no activity at the highest concentration tested

## Data Availability

The data presented in this study are available on request from the corresponding author. The data are not publicly available due to their relevance to an ongoing Ph.D. thesis.

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
