# Peer review of "Armeria maritima (Mill.) Willd. Flower Hydromethanolic Extract for Cucurbitaceae Fungal Diseases Control"

_molecules, 2023, doi:10.3390/molecules28093730_

Round 1
Reviewer 1 Report
1.This experiment only relies on gas chromatography-mass spectrometry and the detection of the antifungal activity of the extract to obtain the conclusion of "Armeria maritima (Mill.) Willd. Flower Hydrophobic Extract for Cucurbitaceae Fungal Diseases Control" Is the experimental argument notcomprehensive and complete enough? More data and experiments are needed to demonstrate it.
2.“1. Introduction” The description of the relevant paragraph in the introduction is long and unclear, but be more brief
3.“3.3.2. Comparison with Other Extracts Tested for Cucumber Protection” The description of this part is not clear enough. Should more relevant verification materials be added to demonstrate it fully.
4. English correction: Many sentences are not easy to understand and not scientific, it is better to ask native speaker to modify throughout the manuscript.
5. Some mistakes should be avoided, for example,formatting issues,spaces problem.
Author Response
Comments and Suggestions for Authors
Q1. This experiment only relies on gas chromatography-mass spectrometry and the detection of the antifungal activity of the extract to obtain the conclusion of "Armeria maritima (Mill.) Willd. Flower Hydrophobic Extract for Cucurbitaceae Fungal Diseases Control" Is the experimental argument notcomprehensive and complete enough? More data and experiments are needed to demonstrate it.
Response:
1.- Concerning the first statement made by the reviewer (“This experiment only relies on gas chromatography-mass spectrometry and the detection of the antifungal activity of the extract to obtain the conclusion of "Armeria maritima (Mill.) Willd. Flower Hydrophobic Extract for Cucurbitaceae Fungal Diseases Control”), we consider that the chosen methodology is appropriate and that the statements made are supported by the presented results. A more detailed rebuttal is provided below:
Apropos of the title of the article, in our view, it is representative of the content of the ms., given that it simply indicates the extract used and the application explored in this work.
Regarding the methodology chosen for extract characterization, GC-MS offers high resolution, sensitivity, and reproducibility, as well as advantages related to sample preparation, instrument availability, and cost considerations. It is well suited for the routine analysis of unknowns, and is widely used in the characterization of extracts and essential oils, being generally regarded as a ‘gold standard’. As for the methodology used for antifungal activity investigation, the in vitro procedure followed is the one defined by the European Committee on Antimicrobial Susceptibility Testing (EUCAST), which is among the most popular guidelines used in antifungal susceptibility testing worldwide (together with those defined by the Clinical Laboratory Standards Institute, CLSI). With regard to the ex-situ methodology, the procedure that we have replicated (defined by Onanan et al.) is not standardized, but it is very similar to those used for ex-situ testing on other horticultural crops in many other studies.
About the conclusions, both the one that appears at the end of the extract (“[…] Thus, the reported results support the valorization of A. maritima as a source of biorationals for Cucurbitaceae pathogens protection, suitable for both organic and conventional agriculture.”) and the one at the end of the actual conclusions section (“The results suggest that the extracts of this halophyte could be valorized as biorationals for the protection of cucurbits against certain soil-borne diseases.”) are consistent with the findings of in vitro efficacies better than those of two widely-used conventional fungicides (and better than the third synthetic fungicide against one of the phytopathogens).
2.- With regard to the second statement, in which the reviewer casts doubts about whether the study is comprehensive and complete enough, we would like to bring to the reviewer’s attention that the extract has not only been tested in vitro against six phytopathogens, alone and in combination with chitosan oligomers, but that a direct comparison has been made with the three main constituents identified in the extract (and their conjugate complexes) as well as with three conventional fungicides. Furthermore, ex-situ experiments have been conducted for the phytopathogen (Sclerotinia sclerotiorum) that could be tested in a laboratory (without the need for running experiments in greenhouse cabins of biological containment, to which we do not have access). Please note that we have now improved the ms. by including a comparison with other plant extracts reported in the literature, which further evidences the high efficacy of the reported extract.
3.- Regarding the third comment (“More data and experiments are needed to demonstrate it.”), we agree that, for instance, in vivo experiments for the other five phytopathogens or field studies would be interesting and would provide additional information, but we feel that they fall outside the scope of this first study. We have now acknowledged this limitation of the study and suggested it as a topic for further research in the Discussion section of the revised manuscript, in a new section on “Limitations of the study and further research”. The new paragraph reads: “While the preliminary in vitro and ex-situ results suggest that the proposed COS-A. maritima conjugate complexes have potential as antifungal agents against Cucurbitaceae fungal pathogens, further research is needed to assess their practical applicability for crop protection. Tests with different fungal strains would be required to factor in differences in sensitivity, and field tests should be conducted on various Cucurbitaceae species. Furthermore, the impact of the treatment on other Cucurbitaceae bacterial and fungal pathogens not tested in this study should also be taken into consideration if traditional fungicides are to be replaced with this natural product-based alternative. Additionally, the timing of application, dosage, and other practical aspects such as cost, degradation tolerance, and efficacy of long-term protection should also be carefully evaluated in future studies.“
Q2. “1. Introduction” The description of the relevant paragraph in the introduction is long and unclear, but be more brief
Response: Following the Reviewer’s recommendation, the introduction has been shortened by 25% (from 956 to 724 words) and Figure 1 has been deleted, so that it now fits into the second page of the manuscript.
Q3. “3.3.2. Comparison with Other Extracts Tested for Cucumber Protection” The description of this part is not clear enough. Should more relevant verification materials be added to demonstrate it fully.
Response: The subsection title has been updated, including “ex-situ”, to make it clearer. The first sentence has also been rewritten, and it now reads: “There is a limited amount of research that has investigated the use of natural extracts to inhibit white mold on cucumber ex-situ”, removing any reference to “in vivo” testing (we included it because it was the term used by Onaran et al, but we agree that it may be confusing). As for the request to include more substances for comparison purposes, please kindly note that we have repeated the bibliographical survey in a more thorough way (including other databases apart from WOS and Scopus) and we have not been able to find other studies on natural extracts applied to ex-situ cucumber protection that could allow direct comparisons with our results.
Consequently, to include more activity comparisons, we have opted for creating a new section focused on comparing the in vitro activity of A. maritima extract with those of other plant extracts for each of the phytopathogens under study. The efficacies of the various extracts/essential oils are summarized in a new table, with 27 new references.
Q4. English correction: Many sentences are not easy to understand and not scientific, it is better to ask native speaker to modify throughout the manuscript.
Response: The use of the English language has been carefully checked by a C2-level certificate holder (according to the Common European Framework of Reference (CEFR), a well-educated native English speaker is technically at such C2-level).
Q5. Some mistakes should be avoided, for example,formatting issues,spaces problem.
Response: The text has been thoroughly proofread removing typographic errors, and we have made sure that the formatting of the revised manuscript strictly meets the journal’s template guidelines.
Reviewer 2 Report
The research conducted on the antifungal effects of the coastal rose does have merit to warrant publication. However, there are some minor changes that can be made to the manuscript before publication.
Abstract: Very well written, clear and concise
Introduction: can be re-written to be more concise. As it is now it is a bit lengthy.
Methodology: Well written and clear
Results: Diagrams were well formatted and information provided was accurate and clear.
Conclusion: Accurate
Author Response
Comments and Suggestions for Authors
The research conducted on the antifungal effects of the coastal rose does have merit to warrant publication. However, there are some minor changes that can be made to the manuscript before publication.
Abstract: Very well written, clear and concise
Q1. Introduction: can be re-written to be more concise. As it is now it is a bit lengthy.
Response: The introduction has been shortened by 25% (from 956 to 724 words) and Figure 1 has been deleted, so that it now fits in the second page of the manuscript.
Methodology: Well written and clear
Results: Diagrams were well formatted and information provided was accurate and clear.
Conclusion: Accurate
Reviewer 3 Report
I have gone through the submitted manuscript. It is definitely interesting. But, authors are advised to clear the following points.
1. Recommend to present GCMS chromatogram in the manuscript.
2. Why and how have conjugates showed better anti-fungal action than conventional drugs? Need to explain in discussion part.
Author Response
Comments and Suggestions for Authors
I have gone through the submitted manuscript. It is definitely interesting. But, authors are advised to clear the following points.
Q1. Recommend to present GCMS chromatogram in the manuscript.
Response: The GC-MS chromatogram has been included in the manuscript (Figure 2), as suggested by the reviewer.
Q2. Why and how have conjugates showed better antifungal action than conventional drugs? Need to explain in discussion part.
Response: At this point, without additional in-detail experiments on the mechanism of action, we can only make an educated guess. As for the enhanced activity of the conjugate complexes with COS vs. the non-conjugated extract, it can be hypothesized that it may stem from an enhanced additive fungicidal activity per se (a new paragraph on the antifungal mechanisms of action of COS has been included in the revised text) or by simultaneous action on multiple fungal metabolic sites [ref.], but it may also be due to the fact that chitosan oligomers can increase the solubility and bioavailability of the bioactive compounds present in the extract. This point has been commented on at the end of subsection 3.2.
Concerning the better performance of the natural product vs. the conventional fungicides, the complex mixture of compounds found in plant extracts may be responsible for their better effectiveness, given that these compounds may act synergistically to produce a more potent antifungal effect than synthetic fungicides. A new paragraph has been included at the end of section 3.3.1.
Round 2
Reviewer 1 Report
1. For a long article, the experimental volume is too small and the experimental procedure is too simple.
2. The conclusion statement is too simplistic, suggesting that the authors include some innovative ideas in the conclusion section.
3. Authors should supplement the article in the context of recently published literature. Yes, it is recommended to highlight the novelty of your research content in conjunction with newly published articles in recent years.
4. Some sentences in the article are not fluent. It is recommended that the author improve the article after reading it to improve the quality of the language.
5. Pay attention to the format of references and refer to journal requirements for revision. For example, some provide DOIs and some do not.
Author Response
Q1. For a long article, the experimental volume is too small and the experimental procedure is too simple
Response: As noted in our response to Q1 in the previous peer-review iteration, we insist on the comprehensiveness and completeness of the study, supported by the positive feedback provided by the other two reviewers (who indicated that the methodology is ‘well written and clear’ and that the information provided in the results in ‘accurate and clear’).
Please kindly note that the plant organs have been characterized by infrared spectroscopy; the plant extract has been characterized by GC-MS; the extract has been tested in vitro against six phytopathogens, both alone and in combination with chitosan oligomers, and a direct comparison has been made with the three main constituents identified in the extract (and their conjugate complexes with chitosan oligomers) as well as with three synthetic fungicides. Furthermore, ex-situ experiments have been conducted for the phytopathogen (Sclerotinia sclerotiorum) that could be tested in a laboratory (without the need for running experiments in greenhouse cabins of biological containment, to which we do not have access).
Regarding the suitability of the methodology, as we explained in our previous rebuttal, GC-MS is well suited for the routine analysis of unknowns and is widely used in the characterization of extracts and essential oils, being generally regarded as a ‘gold standard’ (given that it offers high resolution, sensitivity, and reproducibility, as well as advantages related to sample preparation, instrument availability, and cost considerations). As for the methodology used for antifungal activity investigation, the in vitro procedure followed is the one defined by the European Committee on Antimicrobial Susceptibility Testing (EUCAST). With regard to the ex-situ methodology, the procedure that we have replicated (defined by Onanan et al.) is very similar to those used for ex-situ testing on other horticultural crops in other studies.
The statements made above on the merit of the manuscript can be supported by comparing it with previous works by the same group on other plant species (see https://orcid.org/0000-0003-2713-2786), with a similar experimental procedure and comparable or smaller experimental volume. These works have been published in journals with higher impact factors and better journal rank (Q1 instead of Q2). For example, in a recent article published in the International Journal of Molecular Sciences (2023, 24(2), 1154, https://doi.org/10.3390/ijms24021154), with IF=6.2, only the extract and three of its constituents were assessed (instead of extract, three constituents and their respective conjugate complexes), and only three pathogens were studied (instead of six).
Q2. The conclusion statement is too simplistic, suggesting that the authors include some innovative ideas in the conclusion section.
Response: We believe that the conclusion section meets the usual requirements, as outlined in Nature’s training resources (available at https://www.nature.com/scitable/topicpage/scientific-papers-13815490/), which state that the conclusion should interpret the findings at a higher level of abstraction than the discussion and relate them to the motivation stated in the introduction. Other reviewers have also found the conclusion section to be accurate. However, in response to the Reviewer's query, we have added a sentence at the end of the conclusion that outlines the need for further research (in line with the comments made in subsection 3.4 on Limitations of the Study and Further Research): “[…] However, further studies are needed to assess the impact of the proposed treatment on other Cucurbitaceae pathogens and long-term protection. Additionally, practical aspects for its field application need to be optimized.”
Q3. Authors should supplement the article in the context of recently published literature. Yes, it is recommended to highlight the novelty of your research content in conjunction with newly published articles in recent years.
Response: We would like to bring to the Reviewer’s attention that the article cites 43 articles published in the past five years, including 2 articles from 2023, 8 articles from 2022, 8 articles from 2021, 13 articles from 2020, 4 articles from 2019, and 8 articles from 2018. Therefore, we believe that the article provides sufficient coverage of recent literature (please refer to our response to Q3 in the previous iteration: a new section and 27 new references were included to address that query).
With regard to the request to emphasize the novelty of our study, we have revised the sentence in the third paragraph of the introduction. Instead of indicating that “[…] there is no data on its antifungal activity”, we now specify that “[…] However, there is a lack of information regarding the antifungal and antimicrobial activities of extracts from other plant organs, which indicates a research gap.”. The fourth paragraph was also revised to further highlight the novelty of the study, clarifying that “[…] The study presented herein explored for the first time its potential to protect members of the family Cucurbitaceae […]”.
Q4. Some sentences in the article are not fluent. It is recommended that the author improve the article after reading it to improve the quality of the language.
Response: Although specific indications on which sentences need to be rewritten would have been most appreciated, we would like to emphasize that a C2-level certificate holder carefully checked the English language (see the response to Q4 in the previous iteration). Additional screening with GrammarlyPro and Writefull software tools found no relevant issues. The minor issues detected have been corrected in the revised version. Thus, the manuscript should comply with the journal's quality standards.
Q5. Pay attention to the format of references and refer to journal requirements for revision. For example, some provide DOIs and some do not.
Response: Please note that we managed the references with Endnote software using MDPI’s official template, and details for each of the references had been automatically retrieved from CrossRef or directly downloaded from the publishers’ websites in RIS format (and we made sure that, for instance, scientific names were italicized in the final version). In response to the Reviewer’s request, we have gone through the references in which DOIs were not automatically loaded and added them when available. Specifically, we added DOIs for ref. 4 by Lauranson et al.; ref. 9 by Kumarasamy et al.; ref. 24 by Crespo et al.; ref. 63 by Bergsson et al.; ref. 67 by Jing et al.; and ref. 69 by Yang et al. Additionally, we have included abbreviations for those journals in which the software did not automatically do so.